# Effect of midwife-led pelvic floor muscle training on prolapse symptoms and quality of life in women with pelvic organ prolapse in Ethiopia: A Cluster-randomized controlled trial

Melese Siyoum [1,2]*, Rahel Nardos [3], Biniyam Sirak [4], Theresa Spitznagle [5,6], Wondwosen Teklesilasie [2], Ayalew Astatkie [2]

**1** Department of Midwifery, College of Medicine and Health Sciences, Hawassa University, Hawassa, Ethiopia, **2** School of Public Health, College of Medicine and Health Sciences, Hawassa University, Hawassa, Ethiopia, **3** Department of Obstetrics and Gynecology, and Women's Health, University of Minnesota, Minneapolis, Minnesota, United States of America, **4** Addis Ababa Fistula Hospital, Addis Ababa, Ethiopia, **5** Program in Physical Therapy, Washington University in St Louis, St. Louis, Missouri, United States of America, **6** Worldwide Fistula Fund, Schaumburg, Illinois, United States of America

* melesesiyoum755@gmail.com

## Abstract

### Background

Pelvic organ prolapse (POP) is a common condition that can significantly impact a woman's quality of life. Pelvic floor muscle training (PFMT) is recommended as a first-line conservative treatment for prolapse, but evidence on its effectiveness from low-resource settings is limited. This study aimed to assess the effect of midwife-led PFMT on prolapse symptoms and health-related quality of life (HRQoL) among women with mild-to-moderate POP in Ethiopia.

### Methods and findings

A community-based, parallel-groups, two-arm cluster-randomized controlled trial was conducted in Dale and Wonsho districts of Sidama Region, Ethiopia. Women with symptomatic POP stages I–III were randomized by cluster to receive either midwife-led PFMT plus lifestyle counseling (intervention group) or lifestyle counseling alone (control group). The participants and counselors knew what the women were receiving, but they were not aware of the other group. The outcome assessors, who collected data at the end of intervention, were blinded to the participants' treatment allocation. The primary outcomes were changes in prolapse symptom score (POP-SS) and prolapse quality of life (P-QoL). Mixed-effects generalized linear model was used to determine the effect of PFMT on prolapse symptoms and P-QoL at 99% confidence level. Adjusted β coefficients were used as effect measures. The level of significance was adjusted for multiple comparisons.

**Data availability statement:** All relevant data are within the manuscript and its Supporting information files.

**Funding:** The payment for data collectors, participant transportation, and field supervisor per-diem was sponsored by Hawassa University (HU to MS) (www.hu.edu.et), University of Minnesota (UM to MS) (www.twin-cities.umn.edu), Worldwide Fistula Fund (WFF to MS) (worldwidefistulafund.org) and Maternal Health Fund (MH to MS) (maternalhealthfund.org). The funders had no role in study design, data collection and analysis, decision to publish, or preparation of the manuscript.

**Competing interests:** The authors have declared that no competing interests exist.

**Abbreviations :** CONSORT, Consolidated Standards of Reporting Trials; cRCTs, cluster-randomized controlled trials; HRQoL, health-related quality of life; ICC, intra-cluster correlation; MEGLM, mixed-effects generalized linear model; PFMT, pelvic floor muscle training; POP, pelvic organ prolapse; POP-Q, Pelvic Organ Prolapse Quantification; POP-SS, prolapse symptom score; P-QoL, prolapse-quality of life; PT, physical therapy training; RCT, randomized controlled trial.

A total of 187 women were randomized to intervention ($n = 86$) from four clusters and control ($n = 101$) arms from another four clusters. At sixth months, the intervention group showed significantly greater improvements both in prolapse symptoms and P-QoL. The mean change difference in POP-SS was −4.11 (99% CI [−5.38, −2.83]; $p < 0.001$). Similarly, the mean change difference was: −8.86 (99% CI [−13.84, −3.89]; $p < 0.001$) in physical domain of P-QoL; −11.18 (99% CI [−15.03, −7.32]; $p < 0.001$) in psychological domain of P-QoL, and −9.01 (99% CI [−10.49, −5.54]; $p < 0.001$) in personal relationship domain of P-QoL. A significantly higher proportion (83.72%) of women in the intervention group perceived their condition as "better" after the intervention as compared to 41.58% in the control group. Women with earlier stages of prolapse (stages I and II) experienced higher benefits compared to those in stage III.

## Conclusions

A midwife-led PFMT combined with lifestyle counseling significantly improves prolapse symptoms and quality of life in mild-to-moderate POP. This strategy can be integrated into the existing maternal and reproductive health programs to address POP in low-income settings where access to trained specialist is limited.

## Trial registration

The trial was registered at the Pan African Clinical Trial Registry (https://pactr.samrc.ac.za) database, with the registration number PACTR202302505126575 (https://pactr.samrc.ac.za/TrialDisplay.aspx?TrialID=24311).

## Author summary

### Why was this study done?

- Pelvic organ prolapse (POP) is a prevalent condition that can severely impact women's quality of life, particularly in low-resource settings where access to specialized care is limited.

- Prior to this research, there was limited evidence regarding the effectiveness of pelvic floor muscle training (PFMT) as a conservative treatment option for POP in such contexts, highlighting the need for effective interventions.

### What did the researchers do and find?

- The study utilized a cluster-randomized controlled trial design involving eight clusters (four clusters in each arm) and a total of 187 women, all drawn from these clusters, with mild-to-moderate POP in Sidama region, Ethiopia. The intervention group received midwife-led PFMT combined with lifestyle counseling, while the other group received only lifestyle counseling.

- The key findings from the study showed that women who participated in the midwife-led PFMT combined with lifestyle counseling experienced a significant reduction in their POP symptoms. Specifically, there was an average decrease of 4.11 points in their symptom scores, which indicates a notable improvement in how they felt. Additionally,

the women reported substantial enhancements in various aspects of their quality of life, meaning they felt better overall and were able to engage more fully in daily activities.

**What do these findings mean?**

- The research demonstrates that midwife-led PFMT can serve as an effective first-line treatment for POP, improving symptoms and quality of life in low-resource settings.

- However, limitations include reliance on self-reported data, and a relatively short follow-up period may limit generalizability of the findings.

## Introduction

Symptomatic pelvic organ prolapse (POP) is the descent of one or more of the uterus, cervix, anterior or posterior vaginal wall, or the vaginal apex from its normal position. This descent can be caused by deficient pelvic facia, or weakness in the ligaments or muscles that support the pelvic organs [1]. These structures can be affected by several risk factors including aging, vaginal childbirth, chronic constipation, hereditary connective tissue weakness, persistent elevated intra-abdominal pressure, menopause, and heavy lifting [2].

POP is a prevalent condition affecting many women globally, with estimated prevalence rates varying widely from 1% to 65% [3,4]. This broad range can be attributed to the challenges in accurately estimating prevalence, largely due to the differing methodologies employed in various studies. For instance, researchers may utilize a variety of symptom questionnaires and clinical examination techniques, leading to inconsistencies in the reported prevalence [3,5].

Women with symptomatic POP present with a range of prolapse-related symptoms that are associated with the urinary, sexual, and anorectal symptoms [6]. It is evidenced that depressive symptoms and sexual function disorders are frequently associated with POP [7,8]. In addition to disrupting daily functioning, these multifaceted symptoms of POP have significant negative impact on patients overall quality of life, body image and sexual function [8]. The severity of symptoms is directly related to the degree of prolapse. As the prolapse descends closer to or beyond the hymen, the symptoms become more frequent and bothersome [9,10].

The current POP treatment options are either surgery or conservative management approaches [11]. The conservative management options include pelvic floor muscle training (PFMT), pessary, life-style modification, or watchful waiting, which are recommended as first-line options if the prolapse is not the indication for corrective surgery [2]. Conservative management has the advantage of causing fewer adverse effects and complications [12]. The existing evidence shows that PFMT has a positive effect on prolapse symptoms and severity [13]. Due to its effectiveness and safety, PFMT is indicated as a first-line treatment for women with mild-to-moderate POP [13]. PFMT improves prolapse symptoms by improving pelvic muscle strength, endurance, power, relaxation and combination of these parameters [1].

Awareness of POP and conservative treatment option is very poor among women [14] and healthcare professionals [15] even in high-income countries. To enhance the accessibility and variety of services for women, it is essential to understand various models of service delivery [16]. In low- and middle-income countries, evidence about the feasibility and effect of PFMT on prolapse symptoms and health-related quality of life (HRQoL) is limited.

In Ethiopia, there is a scarcity of evidence on the effectiveness of PFMT, and women often seek healthcare only at advanced stages of prolapse [17] when impact of PFMT may be

limited. Additionally, the number of trained specialists such as urogynecologists or physical therapists are limited in Ethiopia. There are only 450 gynecologists registered in the Ethiopian Society of Obstetricians and Gynecologists out of which only 15 are urogynecologists and 13 are urogynecology fellows in 2021 [18]. In areas of the world with developed physical therapy training (PT) programs, PT management of POP is the gold standard for administering the service [19]. However, it is considered a sub-specialty within the profession and currently, the Ethiopian Ministry of Health does not support specialty practice for the profession of Physical Therapy [20]. Fortunately, midwives, who are generally more accessible than specialists [21], are capable of learning how to deliver PFMT services and are well-positioned to provide this support to women in their communities. To address these challenges and improve access to the conservative management options, research into different models of PFMT service delivery is critical particularly in low-resource settings.

The aim of this study was to assess the effect of PFMT provided by trained midwives on prolapse symptoms and HRQoL among women with mild-to-moderate POP (stages I–III) in Sidama region, Ethiopia. Our hypothesis was that providing midwife-led PFMT approximately monthly alongside three times per-day home-based exercise by participants for 6 months will improve prolapse symptoms and prolapse-quality of life (P-QoL).

## Methodology

### The Consolidated Standards of Reporting Trials (CONSORT) checklist

The Consolidated Standards of Reporting Trials (CONSORT) 2010 statement [22] was used to guide the reporting of this cluster-randomized controlled trial (cRCT). The CONSORT checklist consists of a 25-item checklist that focuses on the key elements for reporting the design, conduct, analysis, and interpretation of RCTs. An extension to the CONSORT statement for reporting cRCTs [23] was also consulted. We have addressed all applicable items from the CONSORT checklist in the reporting of this cRCT (S1 File).

### Study design and participants

A community-based, parallel-groups, two-arm, cRCT was conducted among women with symptomatic POP stages I–III in D-W HDSS. First, a baseline survey was conducted to determine the burden and risk factors of POP and to identify eligible cases [24]. Women aged over 18 years with at least one symptom of POP, confirmed as stages I–III using the standard POP-Q measure [25], were included in the study. The identified cases were then randomized by community clusters into intervention and control arms. Cluster randomization was preferred to minimize spillover of the intervention to the control arm and due to the feasibility of providing the intervention at group level (nearby health facility). The clusters were *kebeles* which are the smallest administrative units in Ethiopia, similar to neighborhoods or wards. They have a wide range of population sizes, from 500 to several thousand residents [26]. The surface area of *kebeles* varies depending on their location and population density, with those in densely populated urban areas being smaller compared to rural kebeles. Each *kebele* is further divided into smaller units called *gots* or *villages*. *Kebele* administrations are responsible for providing basic services, such as health, education, and infrastructure, to their residents. They also play a crucial role in population-based surveys and research studies in Ethiopia, often serving as the primary sampling units [27,28]. Kebeles are organized into larger administrative divisions known as woredas (Districts).

Sample size calculations were performed using Open-Epi version 3.1, based on the following assumptions: a 99% confidence level, 80% power, an exposed-to-unexposed ratio of 1, a pelvic organ prolapse symptom score (POP-SS) mean difference of 3.16 with a standard

deviation of 4.78 in the intervention group, and a mean difference of 0.12 with a standard deviation of 3.86 in the control group over six months [15]. This calculation yielded a required sample size of 96, adjusted for an anticipated 16% refusal and loss to follow-up [15], resulting in an individual-based total sample size of 111.34. This sample size was utilized for all outcomes since we were unable to determine the effect of PFMT on quality of life as measured by the Prolapse Quality of Life Questionnaire. The decision to adjust the confidence level to 99% was based on the adjusted level of significance to 0.0125 than the usual $P$-value of 0.05 to account for multiple comparisons across various outcomes (POP-SS and three distinct domains of P-QoL).

To determine the minimum number of clusters required, the individual-based sample size was multiplied by the intra-cluster correlation (ICC) coefficient. Since there was no reported ICC from earlier studies, we used an ICC of 0.03, which was suggested as an estimated fairly typical value of the ICC [29,30]. This calculation indicated a need for 1.67 clusters per arm. To ensure adequate power [31,32], a total of eight clusters were selected randomly from the 12 available clusters in the D-W HDSS.

After adjusting for cluster effects with a variance inflation factor of 1.39, the total effective sample size increased to 155, resulting in an expected average of 19.4 participants per cluster. However, during enrollment, a slightly varying number of eligible cases was identified in each cluster. Excluding additional participants was deemed unethical, and alternative treatment options could not be provided due to the cRCT design. Furthermore, it was believed that including all eligible cases would be manageable and enhance the study's power [29,30]. Consequently, all eligible cases were included, raising the total sample size from 155 to 187 participants. The exclusion criteria were: women who were diagnosed to have symptomatic POP stages I–III, but planned to leave the study area within 6 months (study period), severely ill women, women with psychiatry disorders, women with delivered myoma, or women who want to have prolapse surgery within 6 months.

## Randomization and masking

The study site, D-W HDSS, comprises eight kebeles from Dale district and four kebeles from Wonsho district. These were stratified into two strata based on their geographical location (Dale and Wonsho districts). Stratifying on location reduces stratum variability and helps to ensure that they are balanced across study arms [31,32]. To improve similarity across treatment and control groups, a balanced number of clusters from each stratum were assigned to each arm. First, all eligible clusters in each district were listed and assigned a unique identifier number. Then, a lottery method was used to select two of four clusters from the Wonsho district and randomly assigned to each arm. Similarly, six of eight clusters from Dale districts were selected and three clusters were assigned to each arm randomly by lottery method. Then, 102 households were selected from each cluster using computer-generated random numbers (based on the list of their house numbers) to identify eligible women. One woman (usually the head of the household) was recruited for the study.

To reduce the risk of spillover-effect of the intervention, the interventional and control clusters were separated by a buffer zone [31], and when it is impossible due to the limited number of clusters in the D-W HDSS, adjacent clusters were assigned to the same arm. Finally, all eligible study participants from each cluster identified during the baseline survey were included in the study (Fig 1).

The enrollment period was open for 8 months from 13 March to 28 October 2023. Each participant was followed for 6 months after enrollment. The last participants were enrolled on October 28, 2023 so the final follow-up visit was on April 27, 2024. Allocation concealment was conducted by staff of D-W HDSS at Hawassa University. To ensure unbiased allocation,

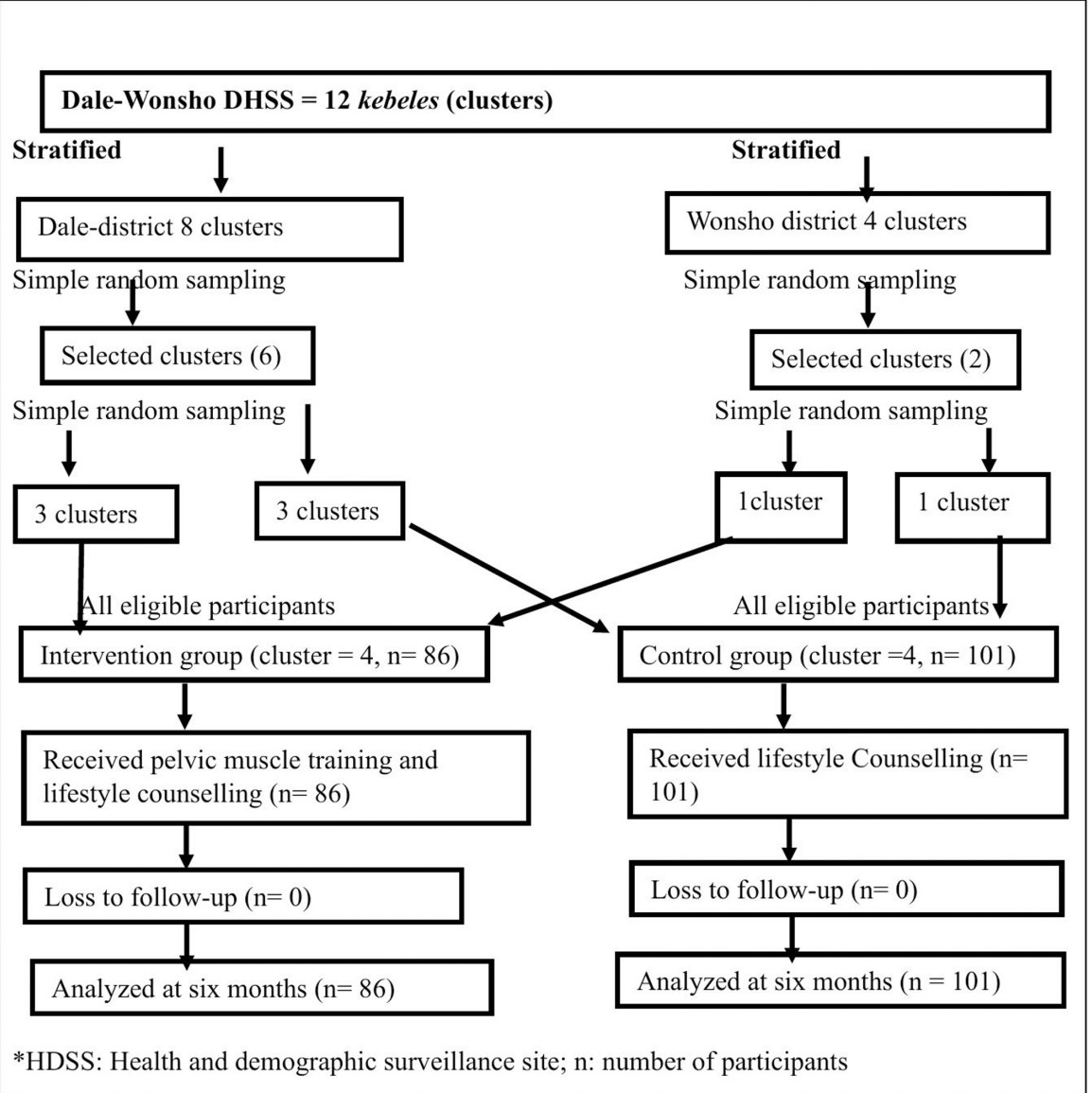

**Fig 1. Randomization and Sampling procedure used to assess the effect of pelvic floor muscle training on prolapse symptoms and quality of life in Sidama Region, Ethiopia, August 2024.**

the lottery method was performed by an independent researcher who was not involved in the recruitment, intervention, or outcome assessment processes. Two staff members of Yirgalem Hamline Fistula Center, who performed pelvic examination to determine the POP stage, recruited eligible participants for intervention.

The participants in the intervention group were aware that they were receiving the PFMT program, but they were not aware of the control group. Similarly, the control group knew that they are receiving lifestyle counseling, but they are not aware of the intervention group. The same is true for the healthcare providers delivering the counseling. Moreover, the outcome assessors, who collected data at the end of intervention, were blinded to the participants' treatment allocation.

## Procedure

In Ethiopia, each kebele/cluster is served by its own health post, staffed by health extension workers. Additionally, groups of five or more kebeles are supported by a single health center, staffed by formally trained healthcare professionals, including midwives. For this study, the intervention was implemented at eight distinct health centers. The intervention was implemented at health centers due to their capacity to deliver the training and their geographical accessibility to participants, especially when compared to hospitals. In this study area, all health centers have at least three degree-holding midwives responsible for providing antenatal, delivery, and postnatal care. Each health center is equipped with various rooms, including delivery rooms, cervical cancer screening rooms, and outpatient rooms, which are suitable for conducting the Pelvic Organ Prolapse Quantification (POP-Q) [25] assessments and ensuring confidentiality during individual counseling sessions.

From each selected health facility, two female midwives were trained either on PFMT and how to counsel patients, or only on how to counsel for life style modification based on their pre-allocated health facility. The training that included education on theory, simulation, and video aids was given by a fellowship trained urogynaecologist. Then, the trained midwives administered the intervention to the women assigned in the intervention arms.

Women who were assigned to intervention received PFMT and prolapse lifestyle advice focusing on risk factors for POP such as constipation, chronic coughing, heavy lifting, and other medical conditions. The control group received only lifestyle modification. In the interventional group, the pelvic examiners link eligible women to the assigned midwife with the POP-Q findings. Then, at the initial visit, women were informed by midwives about the stage of POP they had and type of prolapse (anterior, posterior, or central) which was explained in a private room. Then, the advantage of pelvic floor muscle exercise, and how it works, the detail of the exercise and follow-up schedule were explained in detail at the group level in a private room for 40–60 min.

They were trained to sit or lie comfortably, train pelvic muscles with 10 times 10-s maximum holds, and 10 fast contractions [15,33]. They were instructed to repeat this at least three times per day (morning, afternoon, and night) [15,34]. Women were taught how to precontract the pelvic floor muscles against increases in intra-abdominal pressure (Knack exercise). They were trained to exercise 10 long squeezes and lift contractions, each held for 10 s, allowing time between the squeeze to relax and lower the muscle for up to 5 s. The long squeezes were followed by 10 short squeezes. Women were advised to try to feel the releasing and relaxing of the squeeze after 10 short squeezes, and finally, to do a squeeze with a light cough for 10 times. The long squeezes were for 10 s each and the 10 quick squeezes for 1–2 s each.

To help participants correctly identify and contract their pelvic floor muscles, the participants in the intervention group received PFMT with internal digital biofeedback administered by the trained midwives. During the individual training sessions, midwives used various verbal cues and analogies. For example, participants were instructed to "imagine you are trying to stop the flow of urine or hold back gas. Squeeze as though you are trying to slow down or stop your urine mid-stream" [35]. This relevant analogy helped participants visualize the specific action they needed to perform. Throughout the exercises, midwives provided descriptive feedback to guide participants. They would say things like, "As you squeeze, focus on lifting the muscles around your vagina and anus upwards. You should feel a tightening and lifting sensation" [33,35,36]. This encouraged participants to be aware of the specific muscles being engaged during the contractions. We translated the Lifestyle Advice Leaflet from the POPPY trial [15] Into Sidaamu Afoo, the local language, to enhance understanding and effectiveness during counseling sessions.

They were appointed to attend five visits (at weeks 0, 2, 6, 10, and 15) at a nearby health facility. The appointment frequency was adapted from the United Kingdom National Health Service (NHS) [37]. Initial appointments were scheduled closely together to reinforce correct exercise techniques and ensure participants fully understood all the advice provided. Subsequently, appointments were spaced further apart to promote independent home exercise. A 15-week intensive training was recommended to get adequate muscle strength [38]. The change in prolapse symptoms and HRQoL were analyzed at 6 months.

A lifestyle counseling was given for both arms focusing on the following points [15,36].

**Bowel health.** To avoid constipation, which increases strain on pelvic floor muscles and can exacerbate prolapse symptoms, it's important to consume a balanced diet rich in fruits, vegetables, and adequate fiber. Aim for 1.5–2 l of water daily to maintain bowel regularity. When using the toilet, sit fully rather than hovering, as this promotes muscle relaxation. Position yourself comfortably with your feet apart and hands resting on your thighs or supported. Keep your abdominal muscles relaxed; if you need to bear down, take a deep breath to ease tension. Light pressure on the vaginal bulge towards the back passage may assist with complete bowel emptying.

**Bladder health.** It's crucial to urinate only when the bladder signals the need, rather than going "just in case." Sitting fully on the toilet aids in muscle relaxation, and leaning forward slightly after urination can help ensure complete bladder emptying. Avoid straining, as this can worsen prolapse symptoms. Maintaining the recommended fluid intake of 1.5–2 l per day is essential; reducing fluids does not necessarily decrease urinary frequency and may lead to bladder issues.

**Weight management.** Maintaining a healthy weight is vital, as excess weight can place additional pressure on the pelvic floor. Weight loss may improve prolapse symptoms.

**Reducing intra-abdominal pressure.** For those with a chronic cough, follow prescribed treatments and stay hydrated to prevent strain from coughing. Proper lifting techniques are also important; adopt good habits even for lighter items, tightening pelvic floor muscles before lifting. When lifting heavy loads, divide the weight and lift from low positions to minimize strain.

**Exercise and physical activity.** It is advisable to avoid activities that worsen symptoms, such as prolonged standing. Try to alternate standing tasks with sitting breaks. Additionally, high-impact exercises, like jumping, jogging, or aerobics that involve both feet leaving the ground simultaneously, should be avoided. Instead, opt for low-impact activities like modified aerobics, cycling, using an elliptical trainer, or brisk walking.

## Outcomes

Our primary outcomes were POP symptoms and HRQoL. The POP-SS was measured by the *Sidaamu Afoo* version of the POP-SS questionnaire [39]. The psychometric properties of the questionnaire have been evaluated in the same setting and found to have acceptable reliability and validity. It is a patient-reported-outcome measure that contains seven items with five Likert-scale ranging from 0 to 4, with a possible total score of 0–28 (the higher score representing worse symptom) [40]. HRQoL was measured by using the prolapse quality of life (P-QoL) questionnaire translated from the Amharic version [41]. While the original version has 20 items with nine domains [42], in a previous work designed to adapt the P-QoL locally (*to Amharic version*), the dimensions have been condensed into three: physical function (including General health perception, prolapse impact, role, physical and social limitation, and severity measures); psychological function (including emotional disturbances and sleep/energy disturbances); and personal relationships. Each domain is related to a particular aspect of QoL, and scores in each domain range from 0 to 100, the higher score indicating poor quality of life. This tool has previously been validated in a similar context (northern Ethiopia).

The secondary outcome, women's perceived change In prolapse, was evaluated using the patient global impression of change index (PGI-I), a seven-point Likert scale (1 = very much better to 7 = very much worse) [43]. In this study, patients were classified as improved if they scored 1, 2, or 3 on the PGI-I scale, no change if they scored 4, and worse if they scored 5, 6 or 7.

Intervention adherence was measured in terms of attendance at appointments [15] and the amount of exercise women reported at follow-up time. Women's adherence to training protocol is considered adequate when they complete at least 80% of the training sessions given by midwives [44].

The details of the baseline data collection procedures and tools have been published elsewhere [24]. All tools were translated from English and Amharic into the local language (*Sidaamu Afoo*).

The baseline data about sociodemographic characteristics and prolapse symptoms were collected through face-to-face interviews with women at their homes by eight first-degree nurses using electronic data collection forms prepared using Kobo tool box. The women were subsequently invited for pelvic examination in a nearby health center to confirm their prolapse status and degree of prolapse by using the POP-Q [25]. The POP-Q was performed by staff members of Yirgalem Hamlin Fistula Center who have special training on Fistula and POP identification. Then, women with symptomatic POP stages I–III were interviewed about how much the prolapse had affected their quality of life using P-QoL tool. The detailed POP-Q procedures and data quality assurance method were published elsewhere [24].

At 3 and 6 months of intervention, four first-degree nurses went to the house of the study participants and collected the data through face-to-face interviews. The data collection tool included the POP-SS and P-QoL questionnaires used at baseline, PGI-I change, and questions used to assess their adherence to training protocol [44].

## Statistical analysis

All data analyses accounted for the complex sampling design, which included stratification by district (Dale and Wonsho), clustering by kebele, and weighting for unequal probability of selection. The weighting variable was the product of cluster and household selection weights, normalized by dividing each unit's survey weight by the unweighted average of the survey weights. Categorical variables were summarized as frequencies and percentages, while continuous variables were summarized using means and standard deviations or medians and interquartile ranges.

The analysis was based on the intention-to-treat principle. In practice, all participants adhered to their allocated treatment and completed the follow-up as scheduled. Although the intervention was randomly assigned at the cluster level, outcomes were evaluated at the individual level. An Individual-level analysis was performed using a mixed-effects generalized linear model (MEGLM), incorporating random effects to account for clustering by health centers and survey weights to adjust for unequal selection probabilities.

To complement the primary analysis, a cluster-level analysis was conducted following Hayes and Moulton's framework. Cluster-level means were calculated for key outcomes (mean change in POP-SS, physical domain, psychological domain, and personal relationship) and analyzed using Mann–Whitney U test to compare intervention and control clusters (S5 File).

To compare the primary outcomes at 6 months, we fitted a multilevel MEGLM to the change from baseline in POP-SS and each domain of HRQoL at three and 6 months. The model included fixed effects for treatment group, time, and other relevant covariates (stage of prolapse, duration of prolapse symptoms, residence, parity, age, level of education, history of prolonged labor, wealth index, constipation, and chronic cough), as well as a random intercept for patients within centers and a random slope for time within patients. The relevance of the

covariates was determined based on prior research, clinical aspect, and their potential impact on the intervention and outcomes (prolapse symptoms and quality of life). MEGLM was chosen for its flexibility and robustness needed to accurately model the complex and varied data structures often found in cRCT leading to more reliable and interpretable results [45].

The difference between intervention and control group was estimated by the mean change from baseline at 6 months. Bonferroni correction for multiple measurements [46] was used to adjust the level of significance to 0.0125 as we measured the effect of PFMT on four outcomes (POP-SS and three domains of HRQoL). Then a mean change difference between the intervention and control arms was presented with 99% confidence interval and $P$-values. This adjustment is crucial for maintaining statistical rigor and minimizing the risk of Type I errors in the analysis [46].

To evaluate model fit, we initially fitted a null model for each outcome variable (POP-SS and the three domains of Prolapse Quality of Life (P-QoL)), incorporating random effects for both cluster and participant ID. We calculated the intra-cluster correlation coefficient (ICC) to determine the proportion of variance in the outcome attributable to the clustering structure, which was found to be 4% and less in this study. Subsequently, we fitted models that included the intervention variable to the outcome variables individually. This step allowed us to assess the impact of the intervention on each outcome. We evaluated model fitness using the Pseudo Log-Likelihood values for model fitness and Adjusted Wald Test for significance of predictors. Higher (less negative) Pseudo Log-Likelihood values indicate better model fit and significant Wald test indicates the importance of the predictors [47]. This approach allowed us to identify the most appropriate models for each outcome. Residual analysis was done to assess homoscedasticity and normality of residuals by using the histogram. To check for effect modifiers, we employed statistical tests to assess interactions between PFMT and potential effect modifiers. This approach allowed us to determine whether the effect of PFMT varies across different subgroups. The data set on which the analysis based was provided as a S2 File.

## Results

### Trial profile

A total of 187 eligible women were randomized to the intervention group ($n = 86$) or the control group (101). All of them completed the follow-up and completed the questionnaire at 3 and 6 months. The age of participants ranged from 19 to 80 years with a median of 36 years (IQR: 31–50). All were parous, with a median of 4 childbirths (IQR: 4–6). The median hours of daily heavy lifting work were 4 h (IQR: 3–5). The most common stage of prolapse was stage II 120/187 (64.17%). The median duration of feeling prolapse symptom was 48 (30–60) months with the minimum of 5 months and maximum of 20 years. At baseline, a slightly higher number of women in menopause, family history of prolapse, and history of home birth were observed in the intervention group. The mean baseline POP-SS and quality of life domain scores were almost similar across the groups. Almost half 92 (49.2%) of the participants had prolapse of multiple vaginal compartments; whereby 161 (86.1%) had anterior prolapse, 113 (60.43%) had posterior prolapse and 28/187 (14.97%) had apical prolapse. Overall, while the intervention and control arms were balanced in terms of most characteristics at baseline, there were some important characteristics that showed significant imbalances. Detailed sociodemographic characteristics are presented in Table 1.

### Adherence and follow-up

All study participants adhered strictly to their allocated treatment and completed the follow-up as scheduled. No participants changed their allocation, missed follow-up

**Table 1. Sociodemographic and clinical characteristics of trial participants at Baseline in Sidama Region, Ethiopia (August 2024) ($n = 187$).**

| Variables | Category | Control ($n = 101$) | Intervention ($n = 86$) |
|---|---|---|---|
| Districts | | | |
| Number of clusters | Dale | 3 | 3 |
| | Wonsho | 1 | 1 |
| Number of participants | Dale | 64 | 70 |
| | Wonsho | 37 | 16 |
| Clusters | Aposto | 21 | – |
| | Megara | – | 26 |
| | Dagia | – | 16 |
| | Shoye | – | 27 |
| | Wayicho | 26 | – |
| | Dansha sire | 17 | – |
| | Mamana | 37 | – |
| | Bokaso | – | 16 |
| Age at baseline | Median (IQR) | 34 (30–37) | 50 (35–60) |
| Age at first birth | Median (IQR) | 18 (17–19) | 17 (16–20) |
| Number of childbirths | (Median (IQR) | 4 (4–5) | 5 (4–7) |
| Age at last delivery | Median (IQR) | 30 (26–32) | 36 (29–42) |
| Level of education | No formal education | 67 (66.82%) | 60 (69.47%) |
| | Primary school | 18 (17.94%) | 20 (23.16%) |
| | Secondary and above | 15 (15.25%) | 6 (7.38%) |
| Occupation | Employed | 5 (5.38) | 5 (5.79%) |
| | Un-employed | 96 (94.62%) | 81 (94.21%) |
| Menopause | Yes | 24 (24.22%) | 53 (61%) |
| | No | 77 (75.78%) | 34 (39%) |
| Experience of prolonged labor | Yes | 62 (61%) | 46 (53.16%) |
| | No | 39 (39%) | 40 (46.84%) |
| History of home birth | Yes | 66 (65.47%) | 63 (73.16%) |
| | No | 35 (34.53%) | 23 (26.84%) |
| Family history of prolapse | Yes | 53 (52.47%) | 8 (9%) |
| | No | 48 (47.53%) | 78 (91%) |
| Stage of prolapse | Stage 1 | 11 (11.21%) | 11 (13.16%) |
| | Stage 2 | 65 (64.57%) | 55 (64.21%) |
| | Stage 3 | 24 (24.22%) | 19 (22.63%) |
| Body mass index | <18.5 | 43 (42.15%) | 38 (44.74%) |
| | 18.5–24.9 | 55 (54.26%) | 43 (49.47%) |
| | >=25 | 4 (3.59%) | 5 (5.79%) |
| Type (site) of prolapse | Anterior only | 34 (33.66%) | 37 (43.02%) |
| | Posterior only | 15 (14.85%) | 10 (11.63%) |
| | More than one site | 52 (51.49%) | 39 (45.35%) |
| Mean Base-line POP-SS | Mean, SD | 12.17 ± (4) | 13.86 ± (4.4) |
| **Baseline Symptoms** | | | |
| Protruding mass per vagina | | 97 (96.04%) | 82 (95.35) |
| Vaginal Discomfort/pain | | 97 (96.04%) | 83 (96.51%) |
| Heaviness in the lower abdomen | | 96 (95.05%) | 77 (89.53%) |
| Lower back dragging | | 97 (96.04%) | 79 (91.86%) |
| Strain to urinate | | 85 (84.16%) | 81 (94.19%) |
| Difficulty to empty bladder | | 77 (76.24%) | 81 (94.19%) |

*(Continued)*

**Table 1.** (Continued)

| Variables | Category | Control ($n = 101$) | Intervention ($n = 86$) |
|---|---|---|---|
| Difficulty to empty bowel | | 73 (72.28%) | 67 (77.91%) |
| **Prolapse quality of life (baseline)** | | | |
| Physical function | Median (IQR) | 40.5 (29.7–51) | 43 (35–56.7) |
| Personal relation | Median (IQR) | 44.44 (33–55) | 44.44 (33–55) |
| Psychological function | Median (IQR) | 46.69 (33–53) | 40 (33–53) |

IQR, inter-quartile range; SD, standard deviation; POP-SS, pelvic organ prolapse symptoms score.

appointments, or failed to complete the treatment regimen. Consequently, while our analysis framework was based on the intention-to-treat principle, the actual adherence and follow-up were perfect, ensuring that all participants were analyzed according to their initial allocation.

## Prolapse symptoms and quality of life at 6 months

A summary at the cluster level reveals that all clusters experienced at least a minimum mean change of 1 point in the POP-SS score. Notably, a larger change was observed in the intervention clusters compared to the control clusters, suggesting a more significant impact of the PFMT intervention on participants in those areas (Table 2).

Women in the intervention group reported significant improvements in prolapse symptoms and across all domains of P-QoL at six months. In the unadjusted model, the most substantial improvement was observed in the psychological function component of the P-QoL, with a mean change difference of −11.18 (99% CI [−15.03, −7.32]; $p < 0.001$), indicating a large effect size. Moreover, a significantly higher number of women in the intervention

**Table 2. Change in Pelvic Organ Prolapse Symptom Scores (POP-SS) and prolapse-quality of life domains by cluster after intervention, Sidama Region, Ethiopia, August 2024.**

| Clusters | Group | Post-intervention | Outcome variables | | | |
|---|---|---|---|---|---|---|
| | | | POP-SS | Physical function | Psychological function | Personal relation |
| Aposto ($n = 21$) | Control | Mean ± SD | 11.65 (±3.79) | 41.20 (±19) | 36.25 (±12) | 38.64 (±15) |
| | | Mean difference | −1.13 (±2.39) | −3.64 (±8.78) | −8.12 (±12.08) | −3.86 (±8.18) |
| Bokaso ($n = 16$) | Intervention | Mean ± SD | 8.83 (±4.99) | 35.33 (±14.3) | 38.35 (±12.86) | 29.63 (±11.92) |
| | | Mean difference | −3.17 (±2.32) | −11.93 (±14.24) | −10.00 (±18.22) | −13.89 (±16.61) |
| Dagia ($n = 16$) | Intervention | Mean ± SD | 8.67 (±5.07) | 23.55 (±15.92) | 26.68 (±14.63) | 29.63 (±16.08) |
| | | Mean difference | −6.06 (±3.54) | −19.50 (±11.60) | −11.49 (±18.71) | −16.05 (±15.22) |
| Dansha Sire ($n = 17$) | Control | Mean ± SD | 12.42 (±5.40) | 37.37 (±16.57) | 35.10 (±16.42) | 38 (±19.48) |
| | | Mean difference | −1.16 (±1.82) | −1.42 (±4.56) | −3.16 (±8.02) | −2.92 (±7.44) |
| Mamana ($n = 37$) | Control | Mean ± SD | 9.93 (±4.12) | 39 (±14.97) | 35.33 (±13.78) | 39.09 (±13.61) |
| | | Mean difference | −1.26 (±2.13) | −14.99 (±2.56) | −5.43 (±56) | −3.70 (±6.88) |
| Megara ($n = 26$) | Intervention | Mean ± SD | 7.62 (±5.14) | 23.46 (±14.28) | 29.21 (±13.23) | 32.18 (±18.28) |
| | | Mean difference | −6.0 (±4) | −14.99 (±12.11) | −12.88 (±15.98) | −11.88 (±16.53) |
| Shoye ($n = 27$) | Intervention | Mean ± SD | 8.8 (±5.01) | 33.39 (±13.77) | 30.24 (±14.49) | 28.89 (±12.68) |
| | | Mean difference | −5.90 (±4.77) | −11.61 (±17.69) | −12.45 (±13.67) | −11.11 (±13.52) |
| Wayicho ($n = 26$) | Control | Mean ± SD | 10.41 (±4.26) | 33.24 (±21) | 35.42 (±14.43) | 37.93 (±16.98) |
| | | Mean difference | −1.72 (±3.20) | −3.72 (±6.86) | −8.97 (±12.00) | −4.60 (±11.30) |

SD, standard deviation; POP-SS, Pelvic organ prolapse symptoms score.

group reported their condition as "better" after the intervention compared to the control group. In the control group, more than 27/101 (26%) of women became worse in 6 months (Table 3).

## Effect of covariates on pelvic organ prolapses and quality of life

We observed that women in the stages I and II prolapses experienced more marked improvements compared to stage III prolapse both in prolapse symptom and quality of life. Women with higher POP-SS scores, indicating more severe symptoms, experienced less improvement in the physical domain of quality of life compared to those with lower scores. The analysis showed that as the severity of prolapse symptoms increased, the associated improvements in the physical domain of the Prolapse Quality of Life (P-QoL) questionnaire were diminished. Moreover, after adjusting for the severity of prolapse symptoms and the stage of prolapse, the effect of PFMT on the physical domain of quality of life was slightly diminished but showed increased effects in the personal relationship and psychological domains (Table 4).

**Table 3. Effect of pelvic floor muscle training on prolapse symptom and prolapse quality of life, Sidama Region, Ethiopia, August 2024.**

| Outcome | After 6months | | Change difference from baseline (mean, 99% CI, *P*-value) |
|---|---|---|---|
| | Intervention (*n* = 86) | Control (*n* = 101) | |
| POP-SS (mean ± SD) | 8.4 (±5.15) | 10.8 (±4.48) | |
| Change from baseline (mean ± SD) | −5.44 (±4.03) | −1.34 (±2.47) | − 4.11 (−5.08 −3.14) |
| | *P* < 0.001 | *P* < 0.001[t] | *P* < 0.001[m] |
| **Prolapse-Quality of life** | | | |
| Physical function (mean ± SD) | 28.89 ± (15.6) | 38.1 ± (18.1) | |
| Change from baseline physical function (mean ± SD) | −14.2 ± (14.85) | −2.44 ± (5.68) | − 11.48 (−15.9, −7.1) |
| | *P* <0.001[t] | *P* = 0.0076[t] | *P* <0.001[m] |
| Psychological function (mean, ± SD) | 24.96 ± (10.95) | 35.5 ± (14.33) | |
| Change from baseline psychological function (mean, ± SD) | −17.76 ± (17.46) | −6.52 ± (10.39) | −11.24 (−16.8, −5.7) |
| | *P* < 0.001[t] | *P* < 0.001[t] | *P* = 0.01[m] |
| Personal relation (mean, ± SD) | 30.17 ± (15.13) | 38.5 ± (15.89) | |
| Change from baseline personal relation (mean, ± SD) | −12.81 ± (15.80) | −3.84 ± (8.51) | −8.97 (−13.9, −4.02) |
| | *P* < 0.001[t] | *P* < 0.001[t] | *P* < 0.001[m] |
| **Perceived improvement after intervention** | | | |
| Better | 72 (83.72%) | 42 (41.57%) | 0.0049[x] |
| No change | 10 (11.63%) | 32 (31.68%) | |
| Worse | 4 (4.65%) | 27 (26.73%) | |
| **POP-SS at 6th months** | | | |
| Protruding mass per vagina | 71 (82.56%) | 100 (99.01%) | 0.009[x] |
| Vaginal discomfort/pain | 69 (80.23%) | 100 (99.01%) | 0.005[x] |
| Heaviness in the lower abdomen | 58 (67.44%) | 93 (92.08%) | 0.06[x] |
| Lower back dragging | 64 (74.42%) | 95 (94.06%) | 0.048[x] |
| Strain to urinate | 58 (67.44%) | 81 (80.20%) | 0.2[x] |
| Difficulty to empty bladder | 58 (67.44%) | 70 (69.31%) | 0.46[x] |
| Difficulty to empty bowel | 46 (53.49%) | 67 (66.34%) | 0.19[x] |

CI, confidence interval; SD, standard deviation; POP-SS, pelvic organ prolapse symptoms score; *n*, sample size.

*P*-values presented with [t] indicate paired t test, [m] was based on mixed-effects generalized linear model, and [x] indicate chi-square test.

**Table 4. Adjusted effect of pelvic floor muscle training on prolapse symptoms and quality of life in Sidama Region, Ethiopia, August 2024.**

| Predictors | Category | β(crude) | β(adjusted) | 99%CI | P-value |
|---|---|---|---|---|---|
| **POP-SS** | | | | | |
| Intervention | Mean change | −4.11 | −4.11 | −5.38, −2.83 | <0.001 |
| Stage | Stage I | −5.22 | −4.94 | −7.38, −2.49 | <0.001 |
| | Stage II | −3.03 | −2.99 | −4.83, −1.13 | <0.001 |
| | Stage III | Ref | Ref | | |
| Symptom duration | | 0.0 | 0.03 | 0.005, 0.06 | =0.002 |
| prolong labor | | 1.22 | 0.80 | −0.704, 2.30 | =0.17 |
| Education | Not educated | Ref | Ref | | |
| | Primary school | −0.5 | −0.15 | −1.77, 1.46 | =0.81 |
| | Secondary and above | 0.04 | −0.04 | −2.11, 2.03 | =0.96 |
| **Physical Function** | | | | | |
| Intervention effect | Mean change | − 11.47 | −8.86 | −13.84, −3.89 | <0.001 |
| Stage | Stage I | −39 | −34.9 | −42.16, −27.62 | <0.001 |
| | Stage II | −18.37 | −16 | −22.52, −10.79 | <0.001 |
| | Stage III | Ref | Ref | | |
| Baseline POP-SS | | 1.45 | 0.64 | 0.21, 1.06 | <0.001 |
| **Personal relationship** | | | | | |
| Intervention effect | Mean change | −8.97 | −9.01 | (−10.49, −5.54) | <0.001 |
| Stage of prolapse | Stage I | −24.3 | −23.9 | (−29.65, −18.23) | <0.001 |
| | Stage II | −15.12 | −15.20 | (−20.69, −9.71) | <0.001 |
| | Stage III | Ref | Ref | | |
| Baseline POP-SS | | 0.83 | 0.23 | (−0.08, 0.55) | 0.06 |
| **Psychological function** | | | | | |
| Intervention effect | Mean change | −11.24 | −11.18 | (−15.03, −7.32) | <0.001 |
| Stage of prolapse | Stage I | −18.57 | −16.80 | (−24,34, −9.26) | <0.001 |
| | Stage II | −5.53 | −5.13 | (−9.93, 0.32) | 0.006 |
| | Stage III | Ref | Ref | | |
| Baseline POP-SS | | 0.33 | 0.02 | (−0.43, 0.46) | 0.9 |

POP-SS, pelvic organ prolapses symptom score; CI, confidence interval; Ref, reference category.

The regression table shows a combination of four different regression tables (POP-SS and three domains of Prolapse quality of life).

## Adverse events

No adverse events were observed.

## Discussion

This study showed that midwife-led PFMT combined with lifestyle counseling significantly improves prolapse symptoms with a mean change difference of −4.11 (99% CI [−5.38, −2.83]; $p < 0.001$) in POP-SS and each domain of prolapse quality of life with a mean change difference of −8.86 (99% CI [−13.84, −3.89]; $p < 0.001$) in the physical domain, −9.01 (99% CI [−10.49, −5.54]; $p < 0.001$) in the personal relationship domain, and −11.18 (99% CI [−15.03, −7.32]; $p < 0.001$) in the psychological domain. A significantly higher number of women in the intervention group perceived their condition as "better" since the intervention compared to the control group, highlighting the subjective benefit of PFMT. The change in POP-SS was more noticeable in women with prolapse stage I with a mean change difference of −4.94 (99% CI [−7.38, −2.49]; $p < 0.001$) and stage II with a mean change difference of −2.99 (99%

CI [−4.83, −1.13]; $p < 0.001$) as compared to stage III in reducing POP-SS. Similar effect was observed on each domain of P-QoL. In addition, severity of baseline prolapse symptoms was associated with physical function of the P-QoL domain with a mean change difference of 0.64 (99% CI 0.21, 1.01); $p < 0.001$) compared to minor prolapse symptoms indicating that more severe symptoms were linked to less improvement.

The results of the present study align with existing literature from developed countries indicating the benefit of PFMT for the treatment of POP symptoms and quality of life. A multicenter randomized controlled trial (RCT) study from 25 centers (United Kingdom, Dunedin, New Zealand, and Australia) conducted by Hagen et al. [15] showed that PFMT led to a significant reduction in POP symptoms and improved quality of life. A single-center RCT and systematic review of RCT findings also show that PFMT is effective in reducing POP-symptoms, prolapse stage, and quality of life in women with POP-Q stages I–III [4]. Based on the findings of those studies, the Cochrane reviews [48,49] and national institute for health and care excellence (NICE) guidelines [50] recommended PFMT as a first line of treatment for incontinence and POP, to be provided at primary care facilities [51]. Our finding extends these findings and recommendations to a low-income setting which is of a different cultural and health care context, highlighting the universal applicability of PFMT across diverse settings. Moreover, the observed difference in mean change is statistically significant and clinically important as it exceeded the minimally important change of 1.5 for POP-SS [52]. Again, a remarkable improvement was reported on "a feeling of vaginal discomfort/pain and protruding mass through vagina" which are the major indicators for prolapse surgery and are regarded as the most crucial treatment target [53]. The significant reduction in these key symptoms underscores the importance of PFMT. These two symptoms are closely linked to the physical support provided by the pelvic floor muscles. Strengthening these muscles through PFMT may lead to a more noticeable reduction in these specific symptoms compared to others that may be less directly influenced by muscle strength or may involve more complex physiological changes.

The existing evidence shows that the improvement in POP-SS and P-QoL after PFMT can be attributable to several mechanisms. Overall, PFMT strengthens the pelvic floor muscle, providing better support for pelvic organs which can reduce/stop degree of prolapse and alleviate associated symptoms like vaginal bulging, discomfort, and lower back pain [44,54]. Enhanced muscle strength also improves urinary and bowel control, reducing symptoms like straining to urinate and difficulty emptying the bowel [54,55]. The contraction and relaxation of pelvic floor muscles during PFMT promote blood flow to the pelvic region and stimulate the nerves in the area, improving nerve function [56].

The study indicates that women with stages I and II prolapse experience a greater mean change difference in symptom scores compared to those with stage III, after adjusting for other factors. Borello-France and colleagues reported that women with stage II prolapse were better able to elevate their pelvic floor than those with stage III or IV [57]. Severity of prolapse symptoms was associated with physical function suggesting that more severe POP-SS causes less improvement. After adjusting for prolapse stage and symptom severity, the effect of PFMT remained significant with slightly reduced effect on physical function but slightly increased effects on personal relationship and psychological domains.

The findings of this study have important implications for clinical practice and public health policy in low-resource settings. The first implication is that the result suggests that midwife-led PFMT can be considered as a first-line conservative treatment for mild-to-moderate stage of prolapse in all settings especially in resource-limited settings where access to specialized care is limited. This can be integrated into the existing maternal and reproductive health programs. The second implication is that in settings where the number of urogynecologists/

pelvic physiotherapy experts is limited, like the current Ethiopian context, training midwives to deliver PFMT and lifestyle counseling is feasible and effective to address prolapse symptoms and improve quality of life among women with POP. We recognize that the most recommended form of PFMT is supervised training given by a physiotherapist or urogynecologists who have greater in-depth knowledge [1]. However, this is not feasible in low-resource settings due to shortage of trained specialists (only less than 28 urogynecologists in Ethiopia). It was evidenced that nurse-delivered PFMT have comparable effect with those delivered by specialists when implemented properly [58,59]. In the present study, the observed mean change difference was relatively larger and the compliance/adherence to training was better compared to previous studies. This adherence might be because midwives working in the nearby health facility and the women in the intervention know each other, and that may create trust and acceptability. This is a nice model that can be scaled up in other settings including high-income countries.

This study has its own strengths and limitations. Among others, the key strength of the study includes the rigorous cluster-randomized controlled design which enhances the internal validity of the finding, high adherence rate, and complete follow-up which ensure the reliability of the results. The use of validated and culturally adapted measurement tools ensures the relevance and accuracy of the collected data, and also the use of a complex survey design, which accounts for the clustered nature of the data, enhances the potential generalizability of the results to the target population. Above all, to our knowledge, this study is the first to evaluate the effect of PFMT on prolapse symptoms in Ethiopia.

However, some limitations should be considered. The reliance on self-reported adherence to PFMT and symptom improvement may introduce performance bias since the participants were not blinded to the intervention. These participants' knowledge of their treatment allocation may influence their reported outcomes that could lead to higher treatment effect and reported adherence rate. In addition, the follow-up period was relatively short, which prevents us from determining whether the observed effects are likely to be sustained over time. During randomization, we aimed to balance the two groups by allocating similar clusters using stratified cluster sampling. However, this method cannot account for individual-level variables, which is an inherent limitation of cluster sampling. As a result, the age of the participant, parity, and family history of the participants were not balanced in this study. The age discrepancy may stem from inaccuracies in measuring age, as many individuals in rural Ethiopia have no birth certificates.

## Conclusions

Midwife-led PFMT combined with lifestyle counseling significantly improves prolapse symptoms and quality of life across all domains in women with POP. The observed treatment effects were significant and clinically important. The effects of PFMT were higher in women with earlier stage of prolapse. Moving over, in settings with a shortage of urogynecologists, training midwives to deliver PFMT and lifestyle counseling is a feasible and effective approach to improve prolapse symptoms and quality of life. Future studies should incorporate objective measures of adherence and explore long-term effects. Healthcare organizations should advocate and endorse the training of midwives and other healthcare providers to deliver PFMT. This study was conducted in accordance with the principles of the Declaration of Helsinki. Ethical approval was obtained from Institutional Review Board of Hawassa University, College of Medicine and Health Sciences (Ref. No. IRB/157/14). Written informed consent was obtained with provisions for participants unable to read and write to sign with their index fingers, confirming that the consent had been read to them. Given the sensitive nature of the condition, data regarding self-reported prolapse symptoms were collected in a private location within the participants' homes. Pelvic examinations were conducted in delivery rooms to

ensure participants' physical privacy, while lifestyle counseling and pelvic floor muscle training took place in private rooms. All study information was kept confidential, with participants assigned codes to replace their names and personally identifying information. Access to the key linking these codes to individual identities was restricted to the principal investigator. The risks to participants were minimal. While there was potential for psychological distress related to the pelvic examination, as well as temporary discomfort during physical exams and inconvenience from follow-up visits, these risks were mitigated by allowing participants to withdraw from the study at any time during the consent process. The research team provided travel costs and time compensation, equivalent to approximately four dollars (200 Ethiopian Birr), to the participants. During baseline data collection, around 26 participants with stage IV prolapse were referred to the Yirgalem Hamlin Fistula Center for further management. After the study's completion, women with symptomatic stage III prolapse in the control group were linked with the Yirgalem Hamlin Fistula Center for additional care.

## Supporting information

**S1 File. CONSORT 2010 checklist of information to include when reporting a Randomized trial.**
(DOCX)

**S2 File. The data set on which the analysis based.**
(DTA)

**S3 File. Study protocol.**
(DOCX)

**S4 File. CONSORT for Abstract.**
(DOCX)

**S5 File. Cluster-level analysis of the effects of midwife-led pelvic floor muscle training on prolapse symptoms and prolapse -related quality of life.**
(DOCX)

## Acknowledgments

The study participants and data collectors also deserve gratitude. We also thank Dr. Karen Gold and Joseph Kinahan (Maternal Health Fund), Soja Orlowski (Worlwide Fistula Fund), Katie Brown (University of Minnesota) and Alemu Tamiso (Hawassa University) for their facilitation in transferring funds. Yirgalem Hamlin Fistula Center, Health centers in the study area and Mr. Rekiku Fikre (Hawassa University) deserve acknowledgement for facilitating transportation service support in material.

## Author contributions

**Conceptualization:** Melese Siyoum, Rahel Nardos, Ayalew Astatkie.

**Data curation:** Melese Siyoum, Biniyam Sirak, Theresa Spitznagle, Wondwosen Teklesilasie, Ayalew Astatkie.

**Formal analysis:** Melese Siyoum, Wondwosen Teklesilasie, Ayalew Astatkie.

**Funding acquisition:** Melese Siyoum, Rahel Nardos, Biniyam Sirak.

**Investigation:** Melese Siyoum.

**Methodology:** Melese Siyoum, Rahel Nardos, Biniyam Sirak, Theresa Spitznagle, Wondwosen Teklesilasie, Ayalew Astatkie.

**Project administration:** Melese Siyoum.

**Resources:** Melese Siyoum.

**Software:** Melese Siyoum.

**Supervision:** Melese Siyoum.

**Validation:** Melese Siyoum, Rahel Nardos, Biniyam Sirak, Theresa Spitznagle, Wondwosen Teklesilasie, Ayalew Astatkie.

**Visualization:** Melese Siyoum, Ayalew Astatkie.

**Writing – original draft:** Melese Siyoum, Rahel Nardos, Biniyam Sirak, Theresa Spitznagle, Wondwosen Teklesilasie, Ayalew Astatkie.

**Writing – review & editing:** Melese Siyoum, Rahel Nardos, Biniyam Sirak, Theresa Spitznagle, Wondwosen Teklesilasie, Ayalew Astatkie.

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
