## [Editor Report · Decision Letter 0]

29 Aug 2024

Dear Dr Siyoum, 

Thank you for submitting your manuscript entitled "Effect of Midwife-Led Pelvic Floor Muscle Training on Prolapse Symptoms and Quality of Life in Women with Pelvic Organ Prolapse in Ethiopia: A Cluster-Randomized Controlled Trial" for consideration by PLOS Medicine.

Your manuscript has now been evaluated by the PLOS Medicine editorial staff and I am writing to let you know that we would like to send your submission out for external peer review.

Please re-submit your manuscript within two working days, i.e. by Sep 02 2024 11:59PM.

Kind regards,

Louise Gaynor-Brook, MBBS PhD

Senior Editor

PLOS Medicine

---

## [Decision Letter · Decision Letter 1]

3 Oct 2024

Dear Dr Siyoum,

Many thanks for submitting your manuscript "Effect of Midwife-Led Pelvic Floor Muscle Training on Prolapse Symptoms and Quality of Life in Women with Pelvic Organ Prolapse in Ethiopia: A Cluster-Randomized Controlled Trial" (PMEDICINE-D-24-02632R1) to PLOS Medicine. The paper has been reviewed by subject experts and a statistician; their comments are included below and can also be accessed here: [LINK]

After discussing the paper with the editorial team and an academic editor with relevant expertise, I'm pleased to invite you to revise the paper in response to the reviewers' comments. We plan to send the revised paper to some or all of the original reviewers, and we cannot provide any guarantees at this stage regarding publication.

We ask that you submit your revision by Oct 24 2024 11:59PM. However, if this deadline is not feasible, please contact me by email, and we can discuss a suitable alternative.

Don't hesitate to contact me directly with any questions (lgaynor@plos.org). 

Best regards, 

Louise 

Louise Gaynor-Brook, MBBS PhD 

Senior Editor

PLOS Medicine

lgaynor@plos.org

Comments from the reviewers: 

Reviewer #1: The authors describe results of a cluster randomized trial which compared the effect of pelvic floor muscle training and lifestyle counselling delivered by midwives to lifestyle counselling alone on prolapse-related symptoms and quality of life in Ethiopia. The study design and analysis are generally well done, but there is a lack of clarity and sometimes accuracy when describing some aspects of the methodology and results interpretation. Specific comments follow:

Major points

1. The authors state that the size of the kebeles (in terms of population) can be very variable. Was this taken into account at all in the randomisation/selection of clusters procedures? i.e. how did they ensure that there were roughly equal numbers in each trial arm? Perhaps they could include information on the size of the selected kebeles in the manuscript? More generally, it would be helpful to include cluster-level data in table 1, in addition to the individual-level data.

2. The intervention was implemented at health centres. I found the sentence "Each cluster in the study area is served by its own health posts, staffed by health extension workers, while five or more clusters are supported by a single health center." unclear. Does that mean that a group of 5 of the kebeles included in the study all were covered by the same health centre? If so, then it seems this could lead to issues of contamination, i.e. if women from the intervention arm and the control arm are all receiving care at the same health centre, how did the study team ensure that each woman received the allocated care - how were they identified and how did they ensure there was no contact between women receiving the intervention and those who didn't when they were visiting the health centres? But I think I may be misunderstanding what is meant by "five or more clusters are supported by a single health centre".

3. The authors state "At three months and six months of intervention, four first degree nurses went to the house of the study participants and collected the data through face-to-face interviews. The data collection tool included the POP-SS and P-QoL questionnaires used at baseline, PGI-I change and questions used to assess their adherence to training protocol". How could this last point (adherence to training protocol) be done while keeping the assessors masked to the intervention allocation for each woman, as was earlier stated to be the case in the "masking" section of the methods?

4. The POP-Q measure was used to assess eligibility for inclusion in the study. Did the authors consider assessing POP-Q again at 6 months to provide a more objective endpoint? 

5. In the statistical methods section the authors state that "All data analyses accounted for the complex sampling design, which included stratification by district (Dale and Wonsho), clustering by kebele, and weighting for unequal probability of selection. The weighting variable was the product of cluster and household selection weights…" As far as I can see this is the first mention of any household selection, please could this be clarified?

6. The authors use an individual-level analysis, while accounting for the clustered design. However, this approach could be unreliable with a small number of clusters (as there are in this study). For reassurance, the authors should include an additional analysis which allows for the small number of clusters, for example the cluster-level analysis described in the Hayes and Moulton textbook referenced. It seems that the ICC is fairly low (and indeed, the authors find that there is no improvement to the model fit by allowing for the clustering) so this may be overkill, but it would be reassuring if the same or similar results were found.

7. In the statistical methods the authors state "The model included fixed effects for treatment group, time, and other relevant covariates.." What are the other relevant covariates and how was their relevance determined?

8. The authors describe a rather complicated strategy of analysis approach based on determining the best fit of models containing possible confounders and effect modifiers. I would question whether they need to do this since this is a randomised trial. Were there any potential confounders that were imbalanced in the randomisation and/or that were highly predictive of the outcome(s)? Were the possible effect modifiers pre-specified? Relating to effect modification, please include explanation of how effect modification was tested and provide the results (rather than just saying that there were no significant interactions). 

9. "All prolapse symptoms were significantly improved from the baseline for the intervention group but in the control group, "a feeling of vaginal discomfort/pain and something passing through vagina remained persistent"." This wording is not very clear, as all the symptoms remained persistent in both intervention and control groups, although the proportion was reduced in the intervention group. The wording could be improved.

10. There are some sizeable differences in some characteristics between trial arms, for example age, parity. What happens when these are controlled for in the analysis? The sentence "Overall, the intervention and control arms were balanced in terms of most characteristics at baseline" should be modified to reflect that there are some characteristics (which may be important) which are very imbalanced.

11. The Table 3 title does not accurately reflect what is shown in Table 3. The title suggests that this table will show effect modifiers in some way, but what is actually shown is fixed effect models (i.e. no interaction terms). The title should be changed to reflect what is actually shown. Also in Table 3, it would be good to be clear that it is POP-SS score at baseline that is shown as a covariate in the model results for the quality of life outcomes.

12. The interpretation in the discussion "The effects of PFMT were more noticeable in women with prolapse stage-I with a mean change difference of -5.08 and stage II with a mean change difference of -3.06 as compared to stage III in reducing POP-SS." is not correct. The way this sentence is worded implies that it is the effect of the intervention that differs by stage (i.e. effect modification) but actually what these figures represent are the effect of stage on the improvement (i.e. regardless of trial arm, since that is controlled for as a fixed effect in the model). This needs to be corrected. The sentence "In the present study, women with stage I and II prolapses reported more noticeable improvements compared to those with stage III prolapses indicating that the earlier stage of prolapse are more responsive to PFMT. " suffers from the same issue and should also be re-worded.

13. One potential limitation that should be mentioned in the discussion is that the follow-up is fairly short and so we cannot tell whether the effects are likely to be sustained. This could be commented on.

Minor points

1. In the abstract, results, the authors should specify the number of clusters randomized to each arm (since this is the unit of randomization) as well as the number of women who are in each arm.

2. "Our hypothesis was that providing midwife-led PFMT monthly followed by three times per-day home-based exercise by participants for six months" Should this be "Our hypothesis was that providing midwife-led PFMT monthly alongside three times per-day home-based exercise by participants for six months". Also the midwife-led PFMT is approximately monthly as the first appointments are spaced more closely together.

3. In Table 2, difficulty to empty bowel appears twice, with different data.

4. There is some inconsistency in numbers of decimal places/significant figures presented in tables - it would be good to make everything consistent.

Reviewer #2: 

I have read the first 33 pages of the PDF. It seems that there is a copy in the second half of the document. 

This is a very valuable paper as there is only limited information on prolapse treatment low resource countries. 

Creative manner of blinding that may work well for the purpose of this study. Is there any additional information on the social interaction between the clusters? And this alternative blinding may be listed in the limitations. 

Is 'an abundance of midwifes' the correct wording? (There are still major problems with reaching all women in Ethiopia for midwifery care, as far as I know. But maybe not due to numbers of midwifes but only poor spread over the country?) 

POP-Q may need a reference. 

Page 8: typo in 'participants of for intervention'

Lifestyle intervention might be better explained. This is also an interesting cohort for the effect of lifestyle modification in the control group, but to be useful this needs more explanation and discussion then. There are 4 conditions mentioned but not e.g. what would have been the measures taken in case of constipation, chronic cough etc. or was it only information given, and if yes what type of information. 

Is it known whether the women completed the questionnaires by themselves in their house (fully self-reported)? Were there analphabets included? Or was this done through these face-to-face interviews? We observers/nurses instructed explicitely not to influence the answers? 

This is what is mentioned: 'At baseline, a slightly higher number of women in menopause, family history of prolapse, and history of home birth were observed in the intervention group.' But how do you explain the difference in Age at base line between groups which is unexpected in an RCT? The median in the control group is lower than the lower bound of interquartile range in intervention group. Why was age not considered as effect modifier? 

Cloud you try to explain that 26% of women in the control group became worse in 6 months? 

Reviewer #3: Dear Authors, thank you for allowing me to review your paper and congratulations on conducting your research. I have provided comments under some headings below. My main comment is I believe there are some statistical issues that need addressing.

Methods

Re: "The detail of the counselling was provided using a Sidaamu 249 Afoo (local language) version of a the PROPEL realist evaluation guideline [33]", I am not sure what this means as the PROPEL study was an implementation study relating to a previous trial - POPPY. Did the authors use the Lifestyle Advice Leaflet from the POPPY trial? 

I wondered by the Amharic version of the POP-SS was not used (Belayneh, T., Gebeyehu, A., Adefris, M. et al. Validation of the Amharic version of the Pelvic Organ Prolapse Symptom Score (POP-SS). Int Urogynecol J 30, 149-156 (2019). https://doi.org/10.1007/s00192-018-3825-x) since the Amharic version of the P-QoL was used.

Sample size calculated based on 1% significance level to account for multiple comparisons, however many of the comparisons made were unnecessary. So perhaps 5% level of significance was appropriate. It is not good stats practice to do statistical tests on baseline variables (Table 1). Also I question why do univariate tests (Table 2) when the primary analysis is a multivariable model?

Univariate results

Paragraph from line 392-396: does this relate to Table 2? - add (Table 2) to indicate this. Says "All prolapse symptoms were significantly improved from the baseline for the intervention group but in the control group, "a feeling of vaginal discomfort/pain" and "something passing through vagina" remained persistent" - (extra quotation marks needed) but the reader cannot tell this from Table 2 as there are no pre-post tests presented. In any case I think the pre-post testing is a distraction from the main analysis which is a between groups analysis, and adds to the number of comparisons done.

Modelling methods

Were other baseline variables (eg. age) included in the models? Why are there different moderator variables for different outcomes? What was the process for fitting/retaining/excluding variables from models? In the POP-SS model in Table 3 symptom duration and prolonged labour are included although not significant. Baseline variables are not balanced eg. age, family history. This is not really discussed. It is more important then to include baseline variables in the models to adjust for this.

Line 337: "The random intercept was insignificant and the model without the cluster variable provided a better fit". I query whether this variable should actually be retained in the model regardless as it correctly reflects the cluster trial design.

Paragraph beginning line 400: were interaction terms fitted? This was not mentioned in the methods. If no interactions terms with intervention were included then there was no subgroup analysis. Therefore the statements about the intervention effect being different for different subgroups, eg. stage I and II prolapse, are incorrect. For example the following is not accurate:

"The effects of PFMT were more noticeable in women with prolapse stage-I with a mean change difference of -5.08 and stage II with a mean change difference of -3.06 as compared to stage III in reducing POP-SS." All women with stage I and II had a greater improvement in POP-SS than stage III women (once the effect of the intervention had been accounted for) not just the women in the PFMT arm. In order to test that an interaction term (Stage*Intervention) would need to be included in the model.

Was it baseline POP-SS that was included in the P-QoL models (Table 3). What it the rationale for including an outcome as a predictor variable?

Wording 

Typo in Table 1 and 2 - "Difficulty to empty bowel" appears in 2 lines. One should be "a feeling that your bowel has not emptied completely".

Line 330: "along with the outcome variables" - the intervention variable is fitted to the outcome variable not along with

* Please upload any figures associated with your paper as individual TIF or EPS files with 300dpi resolution at resubmission; please read our figure guidelines for more information on our requirements: http://journals.plos.org/plosmedicine/s/figures. While revising your submission, please upload your figure files to the PACE digital diagnostic tool, https://pacev2.apexcovantage.com/. PACE helps ensure that figures meet PLOS requirements. To use PACE, you must first register as a user. Then, login and navigate to the UPLOAD tab, where you will find detailed instructions on how to use the tool. If you encounter any issues or have any questions when using PACE, please email us at PLOSMedicine@plos.org.

FIGURES AND TABLES

SUPPLEMENTARY MATERIAL

REFERENCES

RCTs

* PLOS Medicine requires that all trials be prospectively registered in one of registries recognized by WHO. Please ensure that study registration details are included in the Methods section.

* Please structure the Methods section using the following sub-headings: Study design and participants, Randomization and masking, Procedures, Outcomes, Statistical analysis.

* Exclusion criteria not detailed in manuscript (from protocol: Women who were diagnosed to have symptomatic Pelvic Organ Prolapse stage I-III but planned to leave the study area within six months (study period), severely ill women, women with psychiatry disorders, women with delivered myoma, or women who want to have prolapse surgery within six months will be excluded.) Please clarify and explain all discrepancies between the paper and protocol. If the outcomes were not prespecified in the protocol, please define them in the Methods (Outcomes section) as post hoc and explain why they were added. Post-hoc comparisons should be presented as hypothesis generating rather than conclusive.

* Please ensure that all prespecified outcomes (primary, secondary, and exploratory) are listed in the Methods/Outcomes section and indicate whether there are outcomes that are not presented in the current report.

* Please specify the dates (Month Day, Year) during which study enrollment and follow up occurred.

* Please include absolute numbers wherever you report percentages; eg, n/N (%)

* Please present the safety data for the study including numbers of specific events and whether or not adverse events are thought to be related to treatment. AEs should be reported in the abstract, per CONSORT and CONSORT-Harms.

* Please complete the CONSORT checklist (https://www.equator-network.org/reporting-guidelines/consort/) and ensure that all components of CONSORT are present in the manuscript, including how randomization was performed, allocation concealment, blinding of intervention, definition of lost to follow-up, power statement. When completing the checklist, please use section and paragraph numbers, rather than page numbers.

* Please report your abstract according to CONSORT for abstracts, following the PLOS Medicine abstract structure (Background, Methods and Findings, Conclusions) https://www.equator-network.org/reporting-guidelines/consort-abstracts/

* If your trial had to undergo important modifications in response to extenuating circumstances, please complete the CONSERVE-CONSORT checklist and provide in your Supporting Information; (https://www.equator-network.org/reporting-guidelines/guidelines-for-reporting-trial-protocols-and-completed-trials-modified-due-to-the-covid-19-pandemic-and-other-extenuating-circumstances-the-conserve-2021-statement/). When completing the checklist, please use section and paragraph numbers, rather than page numbers.

* In keeping with our commitment to Open Science, please include the study protocol document and analysis plan (including any amendments) as Supporting Information to be published with the manuscript if accepted.

* Please note that PLOS Medicine requires prospective, public registration of a data sharing plan (as part of mandatory clinical trials registration) for all clinical trials that began enrollment on or after January 1, 2019, in accordance with ICMJE requirements.

---

## [Decision Letter · Decision Letter 2]

9 Dec 2024

Dear Dr Siyoum,

Many thanks for submitting your manuscript "Effect of Midwife-Led Pelvic Floor Muscle Training on Prolapse Symptoms and Quality of Life in Women with Pelvic Organ Prolapse in Ethiopia: A Cluster-Randomized Controlled Trial" (PMEDICINE-D-24-02632R2) to PLOS Medicine. The paper has been reviewed by subject experts and a statistician; their comments are included below and can also be accessed here: [LINK]

After discussing the paper with the editorial team, I'm pleased to invite you to revise the paper in response to the reviewers' comments. We plan to send the revised paper to some or all of the original reviewers, and we cannot provide any guarantees at this stage regarding publication.

We ask that you submit your revision by Dec 30 2024 11:59PM. However, if this deadline is not feasible, please contact me by email, and we can discuss a suitable alternative.

Don't hesitate to contact me directly with any questions (lgaynor@plos.org). 

Best regards, 

Louise 

Louise Gaynor-Brook, MBBS PhD 

Senior Editor

PLOS Medicine

lgaynor@plos.org

Comments from the reviewers: 

Reviewer #1: The authors have done a reasonable job of responding to the reviewers' comments, but there are still some areas that are not very clear. Specific comments follow:

1. The information that the authors have added to the abstract is useful, but I'm not sure it really needs to be in the abstract (it's a bit distracting). Personally I think it would be ok to just have it in the main text.

2. In the author summary, I suggest including the number of clusters in the first bullet point of the "What did the researchers do and find" section.

3. I still find the paragraph "Each cluster in the study area is served by its own health posts, staffed by health extension workers, while five or more clusters are supported by a single health center. These health centers are staffed by formally trained healthcare professionals, including midwives. The current intervention was implemented at eight different health centers due to their capacity to deliver the training and their geographical accessibility to participants, especially when compared to hospitals." Confusing. It reads as if five of the clusters share the same health centre, but I wonder if it is supposed to mean that there are five cluster (or more?) that are each supported by a health centre, i.e. with different health centres for each of these clusters? I think that what might be causing the confusion is using the term "cluster" (specific to the trial) when actually they mean "kebele" (more general term)?

4. In their response to one of my previous comments, the authors state that "We assigned different data collectors for the intervention and control sites, with two data collectors dedicated to each arm. This separation was intended to minimize any potential bias or contamination during data collection." It is true that this should minimise contamination, but then it could introduce more bias if the data collectors in one arm do things systematically differently to data collectors in the other arm - the authors could comment on this.

5. In their response letter, the authors state that they have used the cluster-level analysis approach (i.e. described in Hayes and Moulton) but there does not seem to be a change to the statistical methods section to reflect this? Also I do not see these results in the tables? (It would be fine to just have them as supplementary tables).

6. With the newly edited sentence "Fortunately, midwives are generally more accessible than specialists, who are potentially capable of learning how to deliver PFMT service" it sounds like it's the specialists who are potentially capable. Suggest re-word to make it clear that the second part of the sentence is talking about the midwives.

7. In the last sentence of the background, suggest the authors replace the word "significantly" by "will".

8. The argument in the discussion that "The age discrepancy may stem from inaccuracies in measuring age, as many individuals in rural Ethiopia has no birth certificates. " isn't very clear - as it would also require that measurement error around age was different in one arm compared to the other trial arm?

Reviewer #2: My suggestions have been addressed well and I agree with the revised version. I like the publication as a significant enhancement of scientific knowledge on POP in Ethiopia.

Reviewer #3: see uploaded comments

* Please upload any figures associated with your paper as individual TIF or EPS files with 300dpi resolution at resubmission; please read our figure guidelines for more information on our requirements: http://journals.plos.org/plosmedicine/s/figures. While revising your submission, please upload your figure files to the PACE digital diagnostic tool, https://pacev2.apexcovantage.com/. PACE helps ensure that figures meet PLOS requirements. To use PACE, you must first register as a user. Then, login and navigate to the UPLOAD tab, where you will find detailed instructions on how to use the tool. If you encounter any issues or have any questions when using PACE, please email us at PLOSMedicine@plos.org.

FIGURES AND TABLES

SUPPLEMENTARY MATERIAL

REFERENCES

STUDY TYPE-SPECIFIC REQUESTS - RCTs 

* PLOS Medicine requires that all trials be prospectively registered in one of registries recognized by WHO. Please ensure that study registration details are included in the Methods section.

* Please structure the Methods section using the following sub-headings: Study design and participants, Randomization and masking, Procedures, Outcomes, Statistical analysis.

* Please ensure that all prespecified outcomes (primary, secondary, and exploratory) are listed in the Methods/Outcomes section and indicate whether there are outcomes that are not presented in the current report.

* Please specify the dates (Month Day, Year) during which study enrolment and follow up occurred.

* Please include absolute numbers wherever you report percentages; eg, n/N (%)

* Please present the safety data for the study including numbers of specific events and whether or not adverse events are thought to be related to treatment. AEs should be reported in the abstract, per CONSORT and CONSORT-Harms.

* Please complete the CONSORT checklist (https://www.equator-network.org/reporting-guidelines/consort/) and ensure that all components of CONSORT are present in the manuscript, including how randomization was performed, allocation concealment, blinding of intervention, definition of lost to follow-up, power statement. When completing the checklist, please use section and paragraph numbers, rather than page numbers.

* Please report your abstract according to CONSORT for abstracts, following the PLOS Medicine abstract structure (Background, Methods and Findings, Conclusions) https://www.equator-network.org/reporting-guidelines/consort-abstracts/

* In keeping with our commitment to Open Science, please include the study protocol document and analysis plan (including any amendments) as Supporting Information to be published with the manuscript if accepted.

* Please note that PLOS Medicine requires prospective, public registration of a data sharing plan (as part of mandatory clinical trials registration) for all clinical trials that began enrollment on or after January 1, 2019, in accordance with ICMJE requirements.

---

## [Decision Letter · Decision Letter 3]

21 Jan 2025

Dear Dr. Siyoum,

Thank you very much for re-submitting your manuscript "Effect of Midwife-Led Pelvic Floor Muscle Training on Prolapse Symptoms and Quality of Life in Women with Pelvic Organ Prolapse in Ethiopia: A Cluster-Randomized Controlled Trial" (PMEDICINE-D-24-02632R3) for review by PLOS Medicine.

I have discussed the paper with my colleagues and it was also seen again by one reviewer who focused on the statistical analysis. I am pleased to say that provided the remaining editorial and production issues are dealt with we are planning to accept the paper for publication in the journal; however, this will be dependent on the statistical reviewer being fully satisfied that their remaining concerns have been addressed - we will consult with them and the Academic Editor on the revision before making a final decision.

[LINK]

We look forward to receiving the revised manuscript by Jan 28 2025 11:59PM.   

Sincerely,

Rebecca Kirk

On behalf of:

Louise Gaynor-Brook, MBBS PhD

Senior Editor 

PLOS Medicine

plosmedicine.org

Requests from Editors:

GENERAL EDITORIAL REQUESTS

* At this stage, we ask that you include a short, non-technical Author Summary of your research to make findings accessible to a wide audience that includes both scientists and non-scientists. The Author Summary should immediately follow the Abstract in your revised manuscript. This text is subject to editorial change and should be distinct from the scientific abstract. Ideally each sub-heading should contain 2-3 single sentence, concise bullet points containing the most salient points from your study. In the final bullet point of ‘What Do These Findings Mean?’ Please include the main limitations of the study in non-technical language.

Please see our author guidelines for more information: https://journals.plos.org/plosmedicine/s/revising-your-manuscript#loc-author-summary."

* Please confirm that your title complies with to PLOS Medicine's style. Your title must be nondeclarative and not a question. It should begin with main concept if possible. "Effect of" should be used only if causality can be inferred, i.e., for an RCT. Please place the study design ("A randomized controlled trial," "A retrospective study," "A modelling study," etc.) in the subtitle (ie, after a colon).

* Please confirm that your abstract complies with our requirements, including providing all the information relevant to this study type https://journals.plos.org/plosmedicine/s/submission-guidelines#loc-abstract

* Please ensure that the Introduction ends with a clear description of the study question or hypothesis.

* Please ensure that all abbreviations are defined at first use throughout the text.

GENERAL

FUNDING STATEMENT

* The funding statement should include: specific grant numbers, initials of authors who received each award, URLs to sponsors’ websites. Also, please state whether any sponsors or funders (other than the named authors) played any role in study design, data collection and analysis, the decision to publish, or preparation of the manuscript. If they had no role in the research, include this sentence: “The funders had no role in study design, data collection and analysis, decision to publish, or preparation of the manuscript.”

COMPETING INTERESTS STATEMENT

* All authors must declare their relevant competing interests per the PLOS policy, which can be seen here: https://journals.plos.org/plosmedicine/s/competing-interests For authors with ties to industry, please indicate whether any of the interests has a financial stake in the results of the current study.

DATA AVAILABILITY

* It appears that you have included data that may breach patient confidentiality including phone numbers and significant dates for the participants. Please edit your data set in accordance with patient consent, and remove any identifying data. For more information on how to share sensitive information please see the guidance here: https://journals.plos.org/plosmedicine/s/data-availability#loc-acceptable-data-access-restrictions

Comments from Reviewers:

Reviewer #1: Thank you for the updates. My only outstanding comment relates to the cluster-level analysis that I requested. I could see from the ChatGPT output that you shared that you were not quite sure about how to go about this. The best approach is to keep it simple. You were correct to use the collapse command to get cluster level summaries for each outcome. The next step in the analysis just involves comparing these results (i.e. a 4 versus 4 data point comparison) using a t-test. There is no need to adjust for other covariates (doing so actually messes up the cluster-level comparison) or to allow for the survey design for example. So you could just generate the cluster-level means and then use a command such as "ttest outcome, by(trialarm)" to do the analysis. I had a quick look at doing this using the figures that you now include in table 2 (which was a very helpful addition). I found that there was still evidence for each of these outcomes of a difference between trial arms, which was reassuring.

Therefore, my only outstanding suggestion is to update the new supplementary table 5 using results from the commands suggested above. Related to this table, I also suggest removing the 4 bullet points underneath, as the cluster-level analysis is widely considered to be a robust approach (more robust than individual level analysis when there are only 8 clusters) - there are many counter-criticisms of individual level analysis methods for 8 clusters that you do not mention (and some of the bullet points included, e.g. number 3, are not really relevant as you are still finding evidence of a difference between trial arms, so clearly lack of power isn't a problem). Ultimately, it is reassuring that both approaches give consistent results.

[LINK]

---

## [Decision Letter · Decision Letter 4]

12 Feb 2025

Dear Dr Siyoum, 

On behalf of my colleagues and the Academic Editor, Annettee Nakimuli, I am pleased to inform you that we have agreed to publish your manuscript "Effect of Midwife-Led Pelvic Floor Muscle Training on Prolapse Symptoms and Quality of Life in Women with Pelvic Organ Prolapse in Ethiopia: A Cluster-Randomized Controlled Trial" (PMEDICINE-D-24-02632R4) in PLOS Medicine.

Before your manuscript can be formally accepted you will need to complete some formatting changes, which you will receive in a follow up email. You will also need to update the manuscript to ensure that the method matches the changes you completed for Table S5. Please be aware that it may take several days for you to receive this email; during this time no action is required by you. Once you have received these formatting requests, please note that your manuscript will not be scheduled for publication until you have made the required changes.

PRESS

Sincerely, 

Rebecca Kirk 

Senior Editor 

PLOS Medicine